# Changes to the Major Trauma Pre-Hospital Emergency Medical System Network before and during the 2019 COVID-19 Pandemic

**DOI:** 10.3390/jcm11226748

**Published:** 2022-11-15

**Authors:** Giuseppe Stirparo, Giuseppe Ristagno, Lorenzo Bellini, Rodolfo Bonora, Andrea Pagliosa, Maurizio Migliari, Aida Andreassi, Carlo Signorelli, Giuseppe Maria Sechi, Nazzareno Fagoni

**Affiliations:** 1Faculty of Medicine, School of Public Health, University of Vita-Salute San Raffaele, Via Olgettina 60, 20090 Milano, Italy; 2Agenzia Regionale Emergenza Urgenza Headquarters (AREU HQ), Via Campanini 6, 20090 Milano, Italy; 3Dipartimento di Fisiopatologia Medico-Chirurgica e dei Trapianti, Via Festa del Perdono 7, 20122 Milano, Italy; 4Fondazione IRCCS Ca’ Granda Ospedale Maggiore Policlinico, 20090 Milano, Italy; 5Dipartimento di Medicina Molecolare e Traslazionale, Università di Brescia, Piazza del Mercato 15, 25121 Brescia, Italy

**Keywords:** major trauma, coronavirus disease 2019, emergency medical service

## Abstract

Objectives: During the coronavirus disease 2019 pandemic, emergency medical services (EMSs) were among the most affected; in fact, there were delays in rescue and changes in time-dependent disease networks. The aim of the study is to understand the impact of COVID-19 on the time-dependent trauma network in the Lombardy region. Methods: A retrospective analysis on major trauma was performed by analysing all records saved in the EmMa database from 1 January 2019 to 31 December 2019 and from 1 January 2020 to 31 December 2020. Age, gender, time to first emergency vehicle on scene and mission duration were collected. Results: In 2020, compared to 2019, there was a reduction in major trauma diagnoses in March and April, during the first lockdown, OR 0.59 (95% CI 0.49–0.70; *p* < 0.0001), and a reduction in road accidents and accidents at work, while injuries related to falls from height and violent events increased. There was no significant increase in the number of deaths in the prehospital setting, OR 1.09 (95% CI 0.73–1.30; *p* = 0.325). Conclusions: The COVID-19 pandemic has changed the epidemiology of major trauma, but in the Lombardy region there was no significant change in mortality in the out-of-hospital setting.

## 1. Introduction

Major trauma is among the leading causes of death in people younger than 45 years, and many factors may influence mortality [1,2]. Increasing the number of emergency departments (EDs) [3,4,5], hospital network optimisation and emergency medical services (EMS) organisation have been reported to improve the outcome of patients with major trauma [6]. The role of EMS in patient survival is closely related to effective triage and the rapid initiation of out-of-hospital rescue procedures [7]. However, during the recent COVID-19 pandemic, the EMS underwent high workload leading to structural changes [8,9], especially for time-dependent medical emergencies requiring rapid intervention as in trauma, but also for stroke, [10,11,12,13] ST-elevated myocardial infarction (STEMI), [14,15,16] and cardiac arrest.

During the 2019 COVID-19 pandemic, ED trauma admissions decreased between 20% and 80% compared to the pre-pandemic phase [17]. This reduction was associated with increases in trauma severity, with critical cases raising from 20% to 40% of all ED trauma admissions [17,18,19,20]. As expected, occupation-related traumas, i.e., car [21] and sport accidents decreased sharply [22,23,24,25]. Giudici et al. [26] found a lower number of road accidents and occupational accidents during the pandemic periods. In addition, they reported longer EMS mission duration and awaiting times for first emergency responses. All these conditions led to a significant increase in deaths at the scene for trauma.

The COVID-19 pandemic rapidly swept Italy and led to the implementation of lockdown on 9 March 2020. More specifically, the Lombardy region and its EMS [27] was among the first to be affected by the pandemic wave [9,28,29,30]. In the Lombardy region, the EMS comprises four headquarters “SOREU” (Stazione Operativa Regionale Emergenze Urgenze) that respond quickly to 112 calls. Each structure is responsible for managing emergencies in specific territories. The responsible health worker answering the emergency call in the dispatch center in SOREU collects and records all information relevant to the emergency scenario in a specific database named EmMa (Emergency Management). Health professionals may identify the scenario as a ‘major trauma’ based on the information received. The scales used to define a major trauma are included in a regional procedure and are either extrapolated from the Revised Trauma Score (RTS) criteria (ref) or based on the dynamics of the event.

The Agenzia Regionale Emergenza Urgenza (Regional Agency for Emergency and Urgency—AREU) rapidly modified the EMS network after the first COVID-19 outbreak to address and overcome problems in rescue mission activation and delivery related to the pandemic period. New standard operative procedures and protocols were implemented for EMS: some aimig to detect the COVID-19 patient at an early stage [26], others to reduce the emergency call pressure on the 112 SOREU for no-COVID19 events. To reduce the duration of EMS missions, new ambulances were deployed to different locations, shortening the time between the first emergency call and the first ambulance dispatched on scene.

The aim of this study was to analyse the impact of the COVID-19 pandemic on the management of major trauma by the Lombardy EMS, by analysing the response time and mortality in the pre-hospital setting between the pandemic period and the previous year.

## 2. Methods

This is a retrospective observational cohort study of the Lombardy region. The study was conducted according to the principles of the Declaration of Helsinki and was approved by the AREU Data Protection Officer in June 2021.

### 2.1. Emergency Medical System in Lombardy

Lombardy is the largest Italian region, with 9.96 million inhabitants and an area of 23,863 km^2^. all missions are coordinated by a single Regional Agency, AREU, which coordinates the rescue through wheeled vehicles or helicopters. AREU’s tecniche processed about 1 million calls in one years and in 800,000 call a veichle was sent to rescue the patient. AREU consist of 265 ambulances with a crew of 2–3 rescuers, 50 Intermediate Rescue Vehicles with a nurse, 59 Advanced Rescue Vehicles and 5 helicopters. Since March 2020, AREU promptly reorganised its activity and increased the number of vehicles in service adding 80 more vehicles for the outbreak response in the Lombardy region.

### 2.2. Data Register

The data were provided by the registry on the website of AREU. The records for “major trauma” in the EmMa database in the years 2019 and 2020 were analysed. The data analysis process was conducted employing the SAS-AREU portal. The portal contains all data related to emergency calls. All scenarios involving major trauma were selected.

Age and gender of patients, time to first emergency vehicle on scene and mission duration were collected. The total mission duration was defined as the time interval starting with the departure of vehicles from the ambulance station and ending with the admission of the patient to the ED.

### 2.3. Statistical Analysis

Categorical variables are presented as number, and continuous variables are presented as median and interquartile range (IQR). Categorical variables were analysed by means of χ^2^ test, and relative odds ratios (OR) with 95% confidence intervals (95% CI) were provided. Proportions of events in different years were assessed using Z-tests. Continuous variables were tested for normality using the Kolmogorov-Smirnov test, and the appropriate analysis was applied for paired data.

Differences were considered significant when *p* < 0.05, otherwise they were considered nonsignificant (NS). Prism 8.0.1 statistical software (GraphPad Software LLC, San Diego, CA, USA) was used for this purpose.

### 2.4. Major Trauma Definition

The Major trauma is defined l:(a)Alteration of vital functions: systolic blood pressure <90 mmHg, alteration of consciousness (Glasgow Coma Scale ~13), respiratory rate >32 or <10 breaths/minute.(b)Anatomy of the lesion: penetrating wounds of the head, neck, chest, abdomen, proximal limbs, elbow or knee; costal volet; clinical suspicion of fracture of the pelvis or of two or more proximal long bones; plegia or paralysis; proximal wrist or ankle amputation; association of traumatic injury with burn ~2.

The SOREU evaluates and monitors high-level trauma situations with particular attention energy: vehicle ejection, death of a vehicle occupant, fall more than two meters, car-pedestrian hit or self-cyclist.

The mortality for major trauma is defined Major trauma mortality is defined as the absence of electrical activity of the heart or absence of breath in which ACLS manoeuvres are no longer useful in resuscitating the patient. Furthermore, the discovery of a patient with injuries incompatible with survival (i.e., decapitation).

## 3. Results

The data collected from the AREU database are shown in Figure 1. A greater number of major trauma diagnosis were observed in 2019 compared to 2020. No difference in the patients’s average age was observed compared (45.2 (20.8) 46.5 (30.0); *p* = 0.10), while more male were reported in 2019 compared to 2020 (74.0 % vs. 70.4%, *p* = 0.009).

In 2020, a significant reduction in the number of major traumas compared to 2019 (2463 vs. 1897; *p* < 0.05) was observed.

Table 1 shows the mean time to first emergency vehicle dispatch to the scene. During the month of March 2020, a peak delay was recorded. Indeed, the time of the first vehicle on scene increased from 11.1 min to 15.7 min (*p* < 0.001). This change coincided with the first wave of the COVID-19 outbreak. The overall mean duration of mission per month is shown in Table 1, comparing 2019 with 2020 data. In March 2020, the overall mean mission duration increases from 71.6 in 2019 to 86.4 (*p* < 0.01). A similar increase in the overall mean mission duration was observed during October (2020 vs. 2019, *p* < 0.05). Consistent with previous observations, this second delay peak was coincident with the second pandemic wave.

In 2020, the average time to the first emergency vehicle dispatched on scene increased from the previous year, i.e., from 12.5 to 14.3 min (*p* < 0.05). The overall average mission duration was also prolonged from 2019 to 2020, i.e., 71.1 vs. 74.5 min (*p* < 0.05). There was a reduction in major trauma diagnoses in March and April, during the first lockdown (OR 0.59 [95% CI 0.49–0.70], *p* < 0.0001).

Mortality in pre-hospital setting increase from 13.0% (326) in 2019 to 14.0% (275) in 2020, but this increase was not significant (OR 1.09 [95% CI 0.73–1.30], *p* = 0.325). Figure 2 shows the mortality trend by month. The largest increase was observed in April, with mortality increasing from 14.8% (29) in 2019 to 23.4% (25) in 2020 but it was not statistically significant (OR 1.74 [95% CI 0.97–3.19], *p* = 0.084). There wasn’t different in mortality rate in gender, (OR 0.80 [95% CI 0.52–1.14], *p* = 0.21).

Table 2 shows the mechanisms of injury and their percentage. In 2020, road accidents and occupational accidents showed a significant reduction, while “Fall from height” and “Violent event” injuries increased.

Table 3 shows the number of diagnoses and deaths for age range in the analysed years. The one-way ANOVA test didn’t show a significant difference between the number of diagnoses and deaths in the two years under scope.

## 4. Discussion

This retrospective analysis showed a reduction in the number of major traumas during the COVID-19 pandemic period in Lombardy region. In 2020, there was a monthly reduction in major trauma compared to the previous year. Nevertheless, overall mortality did not change over the two years.

There are conflicting data in the literature on this topic. Jacob et al. found no significant reduction in major trauma during the lockdown, while Giuntoli M et al. [19] described a 70.9% reduction in hospital orthopaedic admissions over the same period.

In our analysis, mortality referred to the out-of-hospital setting, thus taking into account patients found dead or declared dead at the scene of the trauma. Our study did not evaluate long-term outcomes but, April, while the EMS was under maximum pressure, was the month when major trauma mortality achieved the peak increase. A minimal increase in first response time and overall mission duration was observed. This was more evident during the first wave of the pandemic. Thus, the emergency system in the Lombardy region was able to respond while maintaining standard response times. Nevertheless, if this increase in the duration of the mission was clinically relevant, i.e., 3.3 min in the overall mission duration, remains not clear. Indeed, during the same period the mortality for major trauma increased in a parallel way. Thus, it is likely that even oly a 3 min increase in the EMS mission duration might have played a role in the final outcome, although not statistically significant.

The increase in transportation time observed in our analysis can be explained by two reasons, the first of which is related to the pandemic. The EMS, as pointed out by Valent et al. [9], experienced a massive increase in the number of requests for emergency intervention, which inevitably led to an increase in the duration of EMS interventions. Our results are consistent with the data reported by Giudici et al. [26] The second reason can be traced back to the closure of some EDs, which may have caused an increase in the average transport time without a significant impact on time needed to reach the scene of the accident. Indeed, an important point of the analysis is the reduction in hub-centres for major trauma during the pandemic, decided by the local health authority in the Lombardy region, in order to separate hospitals hosting COVID-19 patients from those not hosting them. Because of this, ambulances transported patients over longer distances during the pandemic, explaining the delays.

During the lockdown, the number of recorded trauma events decreased significantly, as indicated also by Esteban PL et al. [20]. However, little scientific evidence has been produced on the impact of the lockdown on out-of-hospital deaths related to major trauma. Important changes performed in the EMS organization during the pandemic did not cause a significant increase in mortality from major trauma in our large study population. The absence of further peaks in first response time during the subsequent waves of COVID-19 (October–December 2020), was likely the result of new 112 operative procedures and protocols and tools adopted by AREU to counteract the impact of COVID-19 on EMS [31]. In addition, the number of deaths due to major trauma in the second and third pandemic peaks is an important result for all stakeholders involved, as the massive change in the time-dependent diseases network maintained the previous year’s mortality rate.

There wasn’t a difference in the number of diagnoses and deaths by age group in the two years. This is a strange finding because we observed a change in the numerosity of the different major trauma subsets. In particular, we recorded a reduction in the number of accidents at work and road accidents. Conversely, we recorded a percentage increase in falls from height and violent event. Curiously, the percentage increase did not necessarily match an absolute number increase. Indeed, the absolute number of falls from height did not change significantly while the total absolute number of violent events showed a marked increase. We suppose this is due to the fact that violent events increased because of the psychological stress induced by the lockdown measures while fall of heights remained substantially the same since a reduction of workplace falls may have been masked by an increase in falls at home. We conclude that further research on the matter is necessary to better understand the phenomenon.

We hypothesize that such major trauma subsets may be associated with specific age ranges.

Another interesting finding highlighted by our study was the change in the causes of trauma recorded. The reduction in road accidents and occupational accidents is consistent with the findings of Fahy S et al. [17] and related to the lockdown during which people were forced to stay at home. The more than doubled cases of violent injuries observed in 2020 compared to 2019 could be instead related to the emotional stress caused by the pandemic and lockdown condition, as reported by Amerio A. et al. [32].

The strength of the study is represented on the large number of events reported and the area of interest, i.e., Lombardy region which was the one the first and main area hit by COVID-19 breakout in 2020. Limitations were the retrospective design of the study and the different epidemiology of major trauma during the COVID-19 era. Different types of trauma presented different mortality rates, but we did not analyse mortality based on the causes.

## 5. Conclusions

The COVID-19 pandemic changed the epidemiology of major traumas, leading to a reduction in major traumas in 2020, compared to 2019. EMS vehicles were dispatched with a longer time to the trauma scene and the overall rescue mission duration was longer. However, this did not have a clinical impact on pre-hospital mortality. The “Golden Hour” for major traumas did not change significantly. The 112-emergency system was able to ensure adequate transport times and a major trauma mortality rate in line with 2019 standards, underlining the resiliency of the major trauma emergency network.

## Figures and Tables

**Figure 1 jcm-11-06748-f001:**
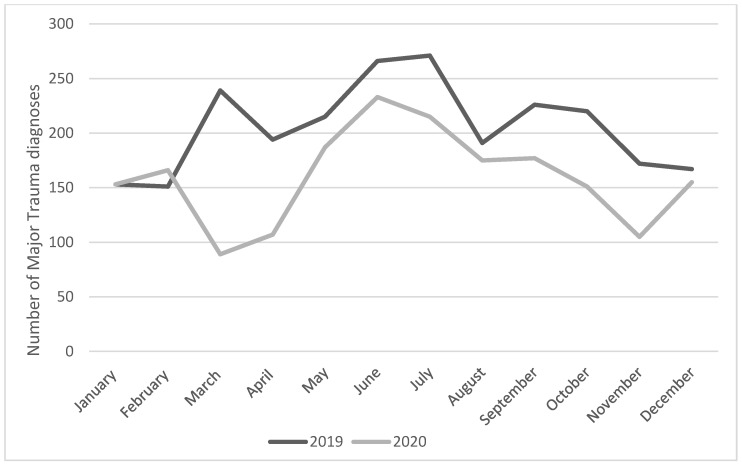
Number of major trauma diagnoses performed by the EMS system.

**Figure 2 jcm-11-06748-f002:**
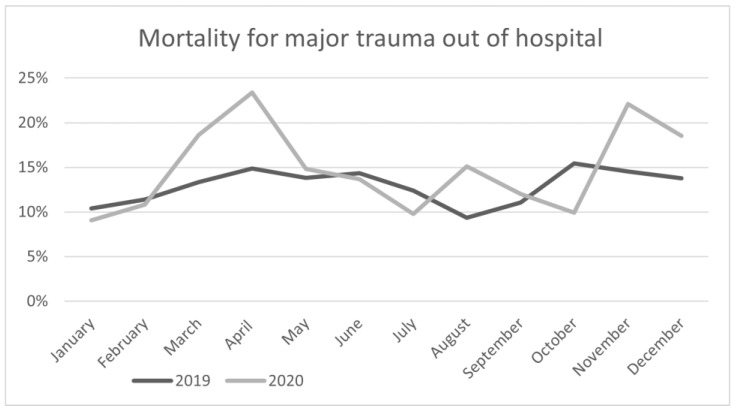
Percentage of deaths from major trauma in the pre-hospital setting.

**Table 1 jcm-11-06748-t001:** Time management. * *p* < 0.05 (*t*-test for paired data). ** time between the call dispatch and the patient’s arrival at the hospital.

	Average Time to First Emergency Vehicle on Scene (Minutes) (SD)		Average Time of Overall Mission Duration (Minutes) (SD) **	
	2019	2020	2019	2020
January	12.2	13.6	70.9	73.2
February	12.8	13.1	71.3	68.9
March	11.1	15.7	71.7	87.0
April	11.1	13.2	69.2	71.8
May	12.1	13.4	70.2	75.5
June	13.4	14.5	68.5	73.4
July	13.4	13.3	72.2	69.0
August	14.2	14.7	73.0	71.5
September	13.0	15.4	70.3	69.5
October	12.3	14.4	69.5	79.3
November	12.5	15.2	71.4	74.4
December	12.3	14.7	74.7	80.6
**Year**	**12.5 ± 0.9**	**14.3 ± 0.9 ***	**71.1 ± 1.7**	**74.5 ± 5.4 ***

**Table 2 jcm-11-06748-t002:** Type of major trauma. “Violent events” are defined as assaults and suicides irrespective of the context, unless otherwise specified (i.e., “fall from height” for some suicides). “accidents at work” are defined as the collective set of all major traumas occurring in the workplace. “Falls from height” are defined as all fall events where the victim falls from a height greater than 1 m.

	Fall from Height		Violent Event		Accidents at Work		Road Accident		Others	
	2019	2020	2019	2020	2019	2020	2019	2020	2019	2020
January	24	16	12	12	23	32	88	76	7	18
February	21	22	8	10	25	26	83	91	12	17
March	20	15	11	11	34	31	162	23	13	6
April	21	17	14	17	23	33	121	34	16	6
May	13	27	16	14	32	39	133	92	23	10
June	25	31	9	27	46	18	170	138	15	20
July	24	18	13	31	36	16	168	143	25	7
August	25	23	12	35	29	6	118	98	8	10
September	22	26	15	26	33	9	134	91	22	23
October	18	24	17	16	33	14	135	87	17	10
November	18	23	9	16	24	12	98	44	23	9
December	18	11	9	24	28	10	103	80	9	26
Total	249	253	145	239	366	246	1513	997	190	162
%	10.1%	13.3%	5.9%	12.9%	14.9%	13.0%	61.4%	52.6%	7.7%	8.5%
*p*	<0.001		<0.001		NS		<0.001		NS	

**Table 3 jcm-11-06748-t003:** Number of diagnoses and number of deaths subset for age range.

	Number of Diagnoses *		Number of Deaths **	
Age Range	2019	2020	2019	2020
1–9	46	34	0	1
10–19	205	150	7	6
20–29	407	288	45	35
30–39	318	233	39	23
40–49	397	291	45	42
50–59	359	297	41	44
60–69	254	212	48	39
70–79	203	167	33	31
80–89	131	103	29	23
90+	17	22	4	7
Missed data	126	100	35	24
Total	2463	1897	326	275

Anova test one way: * *p* value: 0.32; ** *p* value: 0.52.

## Data Availability

The data presented in this study are available on request from the corresponding author. The data are not publicly available in accordance with national data safety guidelines.

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
