# Peer review of "Changes to the Major Trauma Pre-Hospital Emergency Medical System Network before and during the 2019 COVID-19 Pandemic"

_jcm, 2022, doi:10.3390/jcm11226748_

Round 1

Reviewer 1 Report

The article seems correct to me, but they should do a more in-depth study analyzing the variables they study by age and sex and comparing them by the years they analyze. The study they analyze is very basic.

Background.

I think is correct. 

Line 47: delete (refs), they should put the references, delete (refs).

Methods.

The authors should better describe the emergency system of the region they analyze (Lombardia), population, extension, number of transportation resources, type of ambulances….

The authors should specify that they analyze the mortality of the patients and how they define this mortality.

Results.

I think that the results that the authors give us are very basic, I think that they should make an adjustment for age and sex of each one of the results that they offer us and go deeper into the type of traumas that they have analyzed. In fact they show us something in the discussion section but they would have to write it and show it in the results section.

The authors should make a comparison of the traumas observed by age groups and sex. Depending on the years analyzed.

The authors should give us the mean age and standard deviation for each of the years analyzed.

Why do the authors tell us that there are significant differences (p<0.05) between the number of patients assessed in 2020 compared to 2019? Against what other variable are you comparing it?

The authors should adjust the ORs they calculate for age and sex.

The authors could specify the type of traumas that they have analyzed and compared and assess mortality, age, sex and the times they analyze.

I think that if the authors analyze the data by age and sex and adjusting the results, they could go much deeper and discuss more about the results obtained.

Discussion.

The authors should eliminate the results of the first paragraph, this would have to be shown in the results section.

All results from lines 197-204 should be removed and displayed in the results section. To be able to discuss them in the comments section.

Author Response

The article seems correct to me, but they should do a more in-depth study analyzing the variables they study by age and sex and comparing them by the years they analyze. The study they analyze is very basic.

Background.

I think is correct. 

Line 47: delete (refs), they should put the references, delete (refs).

We introduce reference 17.

Methods.

The authors should better describe the emergency system of the region they analyze (Lombardia), population, extension, number of transportation resources, type of ambulances….

We introduce section 2.1 Emergency medical System in Lombardy

The authors should specify that they analyze the mortality of the patients and how they define this mortality.

We introduce section in  2.4 section

Results.

I think that the results that the authors give us are very basic, I think that they should make an adjustment for age and sex of each one of the results that they offer us and go deeper into the type of traumas that they have analyzed. In fact they show us something in the discussion section but they would have to write it and show it in the results section.

The authors should make a comparison of the traumas observed by age groups and sex. Depending on the years analyzed.

The authors should give us the mean age and standard deviation for each of the years analyzed.

We added in raw 130

Why do the authors tell us that there are significant differences (p<0.05) between the number of patients assessed in 2020 compared to 2019? Against what other variable are you comparing it?

We added in raw 136, the reduction is the percentage of diagnoses in 2020 compered to 2019.

The authors should adjust the ORs they calculate for age and sex.

We added the OR for mortality in raw 161.

The authors could specify the type of traumas that they have analyzed and compared and assess mortality, age, sex and the times they analyze.

I think that if the authors analyze the data by age and sex and adjusting the results, they could go much deeper and discuss more about the results obtained.

We introduced table 3 with the results of a one-way ANOVA test aimed to evaluate the eventual difference in the number of diagnoses and deaths in the analyzed years by age range. The results were not significant.

Discussion.

The authors should eliminate the results of the first paragraph, this would have to be shown in the results section. We eliminated the quantitative results from the discussion section.

All results from lines 197-204 should be removed and displayed in the results section. To be able to discuss them in the comments section.

Reviewer 2 Report

This is an interesting study examining changes in pre-hospital emergency medical services and trauma occurrences brought on by COVID-19 in the Lombardy region of Italy. Data were obtained from a well-described database and analyzed to compare 2019 (pre-COVID) to 2020 (during COVID).  The authors present informative charts and tables. The introduction and methods are well-written. However, the methods and results are lacking and the manuscript suffers from inconsistencies in data presentation. The conclusions are not fully supported because of deficiencies in methods.

Strengths:

  • Well-characterized database
  • Interesting area of study
  • Results will be of interest to many EMS units
  • The paper is well-referenced

Weakness

  • The time period under investigation is 2019 compared to 2020. It seems that it should be 01Mar2019-28Feb2020 compared to 01Mar2020-28Feb2021. The inclusion of January and February 2020 in the COVID dataset doesn’t reflect the circumstances going on at the time. The revised time frame would provide a more accurate assessment of the impact of the pandemic.

  • The manuscript would be strengthened by presenting data and statistics on absolute numbers in addition to percent comparisons from year to year. For instance, in Table 2, the numbers on “Fall from height” are essentially unchanged from year to year (249 v 253) but the authors only discuss the change in falls as a percent of the total number of trauma and reach the conclusion that falls increased by a statistically significant amount. This isolation of percentages is misleading. There are likely reasons that the absolute number is the same that should be addressed and discussed.

  • Lines 119-121: it is unclear how the numbers 2463 v 1897 represent a 47.1% decrease.

  • Line 115 regarding males – is the percentage or absolute number?

  • Line 87 – please clarify “age and gender” of whom? The trauma victims?

  • Line 121 says average reduction, and line 156 says average monthly reduction. Is there a difference? Should it be one or the other?

  • The use of the term “pandemics” (plural) is unusual. This is one pandemic with multiple waves.

  • There are many errors in the manuscript which make reading difficult. Figure 1 does not have a y-axis label. Lines 120-122 talk about frequency of trauma in table 1, but table 1 does not show this. Line 129 has numbers that differ from what is shown in the corresponding table. Lin 140 shows numbers that are different from what is shown in the corresponding table. Table 1 should specify “mean” duration. Line 129 should reverse the numbers 74.5 and 71.1 to match the order of the years in the wording right before it. The description of the duration in Line 125 is different from how it is described in the methods on Lines 89 and 90. Numbers in lines 142-143 do not match those in the corresponding table. Line 106 – it is unclear what “mobile thoracic limb” is. Line 135 should refer to Figure 2, not Figure 1.

  • An assessment of multiperson trauma versus individual trauma may be informative but not necessary to the paper.

Author Response

  • The time period under investigation is 2019 compared to 2020. It seems that it should be 01Mar2019-28Feb2020 compared to 01Mar2020-28Feb2021. The inclusion of January and February 2020 in the COVID dataset doesn’t reflect the circumstances going on at the time. The revised time frame would provide a more accurate assessment of the impact of the pandemic.

We address this important point in the study limitation section 

  • The manuscript would be strengthened by presenting data and statistics on absolute numbers in addition to percent comparisons from year to year. For instance, in Table 2, the numbers on “Fall from height” are essentially unchanged from year to year (249 v 253) but the authors only discuss the change in falls as a percent of the total number of trauma and reach the conclusion that falls increased by a statistically significant amount. This isolation of percentages is misleading. There are likely reasons that the absolute number is the same that should be addressed and discussed.

This point was addressed in the discussion.

  • Lines 119-121: it is unclear how the numbers 2463 v 1897 represent a 47.1% decrease.

We apologize for the misunderstanding and typo related to the erroneous interpretation of the following line in the paper results section:

There was a reduction in major trauma diagnoses in March and April, during the first lockdown (OR 0.59 [CI95% 0.49 -0.70], P < 0.0001).

We applied the necessary corrections.

  • Line 115 regarding males – is the percentage or absolute number?

We apologize, it’s a percentage.

  • Line 87 – please clarify “age and gender” of whom? The trauma victims?

We corrected and specified the fact that age and gender are referred to patients.

  • Line 121 says average reduction, and line 156 says average monthly reduction. Is there a difference? Should it be one or the other?

We redesigned

  • The use of the term “pandemics” (plural) is unusual. This is one pandemic with multiple waves.

We changed the “pandemics” terms in “pandemic”.

  • There are many errors in the manuscript which make reading difficult. Figure 1 does not have a y-axis label. Lines 120-122 talk about frequency of trauma in table 1, but table 1 does not show this. Line 129 has numbers that differ from what is shown in the corresponding table. Lin 140 shows numbers that are different from what is shown in the corresponding table. Table 1 should specify “mean” duration. Line 129 should reverse the numbers 74.5 and 71.1 to match the order of the years in the wording right before it. The description of the duration in Line 125 is different from how it is described in the methods on Lines 89 and 90. Numbers in lines 142-143 do not match those in the corresponding table. Line 106 – it is unclear what “mobile thoracic limb” is. Line 135 should refer to Figure 2, not Figure 1.

We fixed all sentences and charts accordingly

  • An assessment of multiperson trauma versus individual trauma may be informative but not necessary to the paper.

Round 2

Reviewer 1 Report

I thank the authors for the effort they have made to improve the manuscript and answer any questions I had.

Reviewer 2 Report

The authors have corrected errors and added content that strengthens the manuscript.